# Machine learning for passive mental health symptom prediction: Generalization across different longitudinal mobile sensing studies

**Daniel A. Adler**[1]*, **Fei Wang**[2], **David C. Mohr**[3], **Tanzeem Choudhury**[1]

**1** Department of Information Science, Cornell Tech, New York, New York, United States of America,
**2** Department of Population Health Sciences, Weill Cornell Medicine, New York, New York, United States of America, **3** Center for Behavioral Intervention Technologies, Department of Preventive Medicine, Northwestern University Feinberg School of Medicine, Chicago, Illinois, United States of America

* dadler@infosci.cornell.edu

**Data Availability Statement:** The two raw datasets used in this work can be found on the Precision Behavioral Health Initiative @ Cornell Tech's website (https://pbh.tech.cornell.edu/data.html). We created a public repository (https://github.com/

## Abstract

Mobile sensing data processed using machine learning models can passively and remotely assess mental health symptoms from the context of patients' lives. Prior work has trained models using data from single longitudinal studies, collected from demographically homogeneous populations, over short time periods, using a single data collection platform or mobile application. The generalizability of model performance across studies has not been assessed. This study presents a first analysis to understand if models trained using combined longitudinal study data to predict mental health symptoms generalize across current publicly available data. We combined data from the CrossCheck (individuals living with schizophrenia) and StudentLife (university students) studies. In addition to assessing generalizability, we explored if personalizing models to align mobile sensing data, and oversampling less-represented severe symptoms, improved model performance. Leave-one-subject-out cross-validation (LOSO-CV) results were reported. Two symptoms (sleep quality and stress) had similar question-response structures across studies and were used as outcomes to explore cross-dataset prediction. Models trained with combined data were more likely to be predictive (significant improvement over predicting training data mean) than models trained with single-study data. Expected model performance improved if the distance between training and validation feature distributions decreased using combined versus single-study data. Personalization aligned each LOSO-CV participant with training data, but only improved predicting CrossCheck stress. Oversampling significantly improved severe symptom classification sensitivity and positive predictive value, but decreased model specificity. Taken together, these results show that machine learning models trained on combined longitudinal study data may generalize across heterogeneous datasets. We encourage researchers to disseminate collected de-identified mobile sensing and mental health symptom data, and further standardize data types collected across studies to enable better assessment of model generalizability.

CornellPACLab/data_heterogeneity) with analysis code. All code written for this study was built using open source Python packages, including sci-kit learn for machine learning, statsmodels for linear modeling, and pingouin for statistical testing.

**Funding:** DCM acknowledges support from NIMH grant R01MH111610. FW acknowledges support from NSF 1750326 and NIA RF1AG072449. TC acknowledges support from the NIH, which supported some of the RCTs discussed within the paper. DA is supported by the National Science Foundation Graduate Research Fellowship under Grant No. DGE-2139899. Any opinions, findings, and conclusions or recommendations expressed in this material are those of the author(s) and do not necessarily reflect the views of the National Science Foundation. This work was funded by a Microsoft Azure Cloud Computing Grant through the Cornell Center for Data Science for Enterprise and Society. The funding sources did not have any role in the study design, collection, analysis, and interpretation of data.

**Competing interests:** DA was co-employed by UnitedHealth Group while conducting this analysis, outside of the submitted work. TC is a co-founder and equity holder of HealthRhythms, Inc., is co-employed by UnitedHealth Group, and has received grants from Click Therapeutics related to digital therapeutics, outside of the submitted work. DA and TC hold pending patent applications related to the cited literature. DCM has accepted honoraria and consulting fees from Apple, Inc., Otsuka Pharmaceuticals, Pear Therapeutics, and the One Mind Foundation, royalties from Oxford Press, and has an ownership interest in Adaptive Health, Inc. FW declares no competing interests. This does not alter our adherence to PLOS ONE policies on sharing data and materials.

## Introduction

Mental health measurement is largely dependent upon patient self-reports, limited to infrequent and inaccessible clinical visits, resulting in delayed treatment. Motivated by these limitations, ubiquitous computing and mental health researchers have explored using near-continuous data streams passively collected from mobile devices (*mobile sensing*) to remotely measure behaviors associated with mental health [1]. Behavioral features are input into machine learning models, which are trained to predict self-reported or clinically-rated symptoms of mental health [2–7]. In addition to mobile sensing data, researchers have explored using brain images, neural activity recordings, electronic health records, voice and video recordings, and social media data to predict mental health outcomes [8–13].

Bardram et al. highlighted that despite the enormous potential of mobile sensing technologies for remote mental health symptom assessment, the field is far from introducing mobile sensing derived measures of mental health in practice, specifically highlighting that the diversity of data types collected across studies creates challenges for cross-study validation, and there is a lack of research into the reproducibility and generalizability of prediction models [14]. To date, most machine learning models leveraging mobile sensing data to predict mental health symptoms have been trained and validated within the context of a single longitudinal study [15–25]. Thus, using these models in practice is tenuous, as symptom-mental health relationships are heterogeneous, and models are not guaranteed to generalize outside of any particular homogenous population [26–28]. Studies often collect data from a single type of device or mobile application [2,4,27,28]. Software and hardware evolve, and these evolutions can change prediction performance [29]. There is a critical gap in the literature to understand if machine learning models trained using heterogeneous datasets containing distinct populations, collected at different time periods, and with different data collection devices and systems, generalize—i.e. models trained using combined retrospective data to predict held-out participants' mental health symptoms across multiple studies achieve similar performance compared to models trained using data collected exclusively from each individual study.

This study addressed this gap by exploring if machine learning models can be trained and validated across multiple mobile sensing longitudinal studies to predict mental health symptoms. We leveraged data from two longitudinal mobile sensing studies: a clinical study of individuals living with schizophrenia, and a non-clinical study of university students. Studies took place 2 years apart, using different mobile applications and smartphone generations to collect data. To the best of our knowledge, the data collected from these studies are the only two examples of publicly available data collected to predict longitudinal mental health symptoms from mobile sensing data thus far. Though the studied populations are very different, both schizophrenia patients and university students exhibit increased levels of depression and anxiety symptoms compared to the general population [30–34]. By analyzing if machine learning models trained by combining data from these two distinct populations generalize, we are not hypothesizing that the psychopathology of schizophrenia patients is similar to college students. Instead, we are exploring if the manifestation of shared mental health symptoms within mobile sensing derived behavioral features between two distinct populations changes a machine learning model's predictive power. It is entirely possible that the relationship between behavior and mental health within the two study populations are too differentiated, and the combined data decreases the model's predictive power. In this paper, we aim to uncover if and when this is true.

Our results show the difficulties of aligning both mobile sensing behavioral features and symptom self-reports across two distinct studies, and we discuss suggestions to improve sensing feature and cross-study symptom alignment, opening the door to continued work

analyzing model generalizability. We then explored if models generalize across symptoms and study populations, and identified a distance metric quantifying the expected model performance improvement as training and held-out validation behavioral feature distribution alignment increased. We experimented with methods to personalize models, and oversampling to improve prediction of severe mental health symptoms underrepresented in data, underpredicted by machine learning models, yet most critical to detect [2,35,36].

## Methods

In this section, we first summarize the StudentLife and CrossCheck studies and data, which are the two longitudinal mobile sensing datasets analyzed in this work. Data collection was not completed in this study, and all analyses included in this study were completed on de-identified publicly released versions of the datasets, downloaded from [37,38]. Please see [3,4] for further details on data collection. We then describe the specific analyses used in this work to explore if models trained using combined (CrossCheck and StudentLife) longitudinal study data to predict mental health symptoms generalize. Specifically, we describe methods used to align collected sensor data and outcome measures across the two datasets, train and validate machine learning models, oversample minority outcomes to reduce class imbalance, and personalize models by aligning behavioral feature distributions. Table 1 summarizes the datasets used in this work and Fig 1 summarizes the modeling flow.

**Table 1. Comparing the CrossCheck and StudentLife datasets used in this work.**

|  | CrossCheck | StudentLife |
|---|---|---|
| **Populations** | Patients with schizophrenia, schizoaffective disorder, or psychosis non-specified in treatment at a hospital in the New York Metro area, with at least one major psychosis-related psychiatric event reported in the past year. | Students at an Ivy League University in the Northeast United States taking a Spring-term Computer Science course. |
| **Intended Study Duration** | 1 year | 10 weeks |
| **Data Collection Period** | 2015–2017 | Spring 2013 |
| **Data Collection Devices** | Samsung Galaxy S5 (Android operating system). | Personal Android phones (devices vary). Those who did not own an Android phone were given a Nexus 4s provided by the researchers. |
| **Number of Participants, n** | 61 | 48 |
| **Age, mean (SD)** | 37.11 (13.85) | Not available in dataset or paper |
| **Female, n (%)** | 25 (41%) | 10 (21%) |
| **Caucasian** | 22 (36%) | 23 (48%) |
| **Asian** | 1 (2%) | 23 (48%) |
| **African American** | 18 (30%) | 2 (4%) |
| **Pacific Islander** | 4 (7%) | 0 (0%) |
| **American Indian/ Alaskan Native** | 1 (2%) | 0 (0%) |
| **Multiracial** | 13 (21%) | 0 (0%) |
| **Missing** | 2 (3%) | 0 (0%) |

Information about CrossCheck and StudentLife was extracted from previous publications [4,39]. SD: Standard deviation.

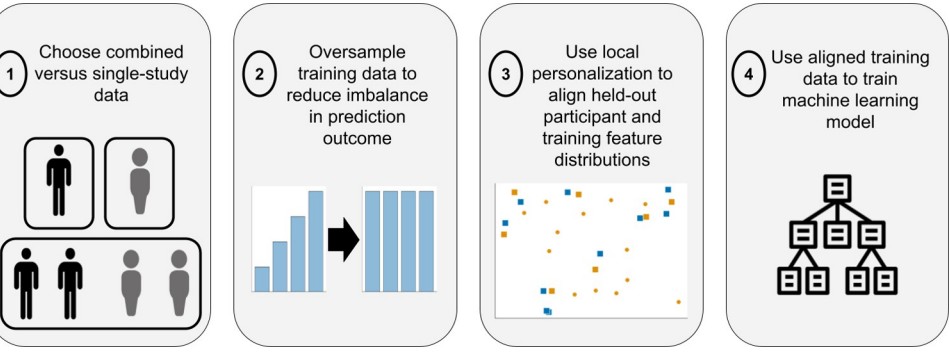

**Fig 1. Modeling overview.**

### CrossCheck study and dataset

Wang et al. collected and partially released data from the smartphone arm of the CrossCheck study publicly [3]. The CrossCheck study was conducted between 2015–17 to develop mobile sensing indicators of schizophrenia symptoms. Adult participants with a medical record diagnosis of schizophrenia, schizoaffective disorder, or psychosis were recruited from treatment programs at a psychiatric hospital in the northeast U.S., could operate a smartphone, had a sixth grade reading level, provided informed consent, and had psychiatric crisis management 12-months prior to study enrollment. Participants were then randomized into the smartphone/non-smartphone arms of the study. This study only used publicly available smartphone sensing data from participants in the smartphone arm of the CrossCheck study. Participants in the smartphone arm were loaned a Samsung Galaxy S5 Android phone. The CrossCheck study was approved by the Dartmouth College and Northwell Health System IRBs, and registered as a clinical trial (NCT01952041) [3].

Participants downloaded and installed the CrossCheck application, which passively collected smartphone sensing data and administered ecological momentary assessments (EMAs) for 12 months. The public CrossCheck dataset is composed of calculated daily and hourly mobile sensing behavioral features and EMAs from 61 individuals. Other surveys, clinical information, and demographic data collected during the CrossCheck study were neither publicly released nor used in this research [2,3,5].

**CrossCheck sensing data.** The CrossCheck application used the Android activity application programming interface (API) to infer if individuals were on foot, still, on a bicycle, tilting, or conducting an unknown activity. Activity was collected every 10 seconds during movement and 30 seconds when stationary. The application tracked conversational episodes (not content) and daily bed/wake times. GPS coordinates were transformed to track unique locations and travel distance. Call and text messaging metadata and the duration and number of times the phone was unlocked were extracted. Lastly, the application tracked ambient noise/light [3].

**CrossCheck EMA data.** The CrossCheck application administered 10 EMAs to participants every Monday, Wednesday, and Friday to track symptoms of schizophrenia, summarized in Table 2 [3]. Participants were asked if they had been feeling depressed, stressed, bothered by voices, visually hallucinating, worried about being harmed, feeling calm, social, sleeping well, could think clearly, and were hopeful. Responses were recorded for each EMA on a scale of 0 (not feeling the symptom at all) to 3 (extremely feeling the symptom).

**StudentLife study and dataset.** The StudentLife study assessed the relationships between smartphone sensing data and mental health outcomes of U.S. college students during the 10-week Spring 2013 term. Participants in a computer programming class were eligible to

**Table 2. The ecological momentary assessment (EMA) symptom outcome measures collected during the Cross-Check study.**

| From 0 (Not at all) to 3 (Extremely)... |
| --- |
| Have you been feeling CALM? |
| Have you been SOCIAL? |
| Have you been bothered by VOICES? |
| Have you been SEEING THINGS other people can't see? |
| Have you been feeling STRESSED? |
| Have you been worried about people trying to HARM you? |
| Have you been SLEEPING well? |
| Have you been able to THINK clearly? |
| Have you been DEPRESSED? |
| Have you been HOPEFUL about the future? |

participate. 48 total participants consented and completed the study. The StudentLife study was approved by the Dartmouth College IRB [4].

Participants were given or used their own Android phone for data collection. Participants downloaded the StudentLife application, which passively collected smartphone sensing data and administered EMAs for 10 weeks. The public StudentLife dataset is composed of raw smartphone sensing, EMAs, and survey data collected from participants. Surveys were administered upon study entry/exit to assess baseline mental health, and educational data was obtained. Corresponding survey and educational data was not available in the CrossCheck dataset and not used in this research.

**StudentLife sensing data.** The StudentLife application automatically inferred whether individuals were walking, running, stationary, or conducting an unknown activity. Conversational episodes (not content) were tracked, as well as WiFi and bluetooth scan logs to determine indoor locations. GPS longitude and latitude coordinates were collected to track outdoor location. The study application extracted call/text logs, duration/number of times the phone was locked for $\geq 1$ hour, and charge duration. The application also inferred when participants were in a dark room for $\geq 1$ hour [4].

**StudentLife EMA data.** Participants were prompted through the application to answer a variety of EMAs. EMAs were administered at varied frequencies, and were occasionally added or removed throughout the study to collect participants' perspectives on specific events. Administered EMAs asked participants about their emotions (e.g. "In the past 15 minutes, I was calm, emotionally stable."), physical activity, mood, current events, sleep, stress, and sociality. In this study, we specifically focused on EMAs that asked students about their mental health, summarized in Table 3.

## Sensor-EMA alignment across studies

We aligned raw StudentLife data to the CrossCheck daily feature data. While publicly released CrossCheck data included daily and hourly features, we used daily features following prior literature analyzing the CrossCheck data to predict triweekly EMAs [3]. The daily data included, for each variable, a daily summary feature and four 6-hour epoch features summarizing data from 12AM-6AM, 6AM-12PM, 12PM-6PM, and 6PM-12AM. For example, for each day, the data included a single feature describing the total number of conversations an individual engaged in throughout a day, and 4 features describing the number of conversations within each 6-hour epoch. We computed the equivalent daily and four 6-hour epoch features for each

**Table 3. The mental health ecological momentary assessment (EMA) symptom outcome measures collected during the StudentLife study.**

| Question |
| --- |
| Do you feel AT ALL happy right now (Yes/No)? |
| Do you feel AT ALL sad right now (Yes/No)? |
| How are you right now? (1) Happy, (2) Stressed, (3) Tired |
| How do you think you will be this time tomorrow? (1) Happy, (2) Stressed, (3) Tired |
| How would you rate your overall sleep last night? (1) Very good, (2) Fairly good, (3) Fairly bad, (4) Very bad |
| Right now, I am. . . (1) A little stressed, (2) Definitely stressed, (3) Stressed out, (4) Feeling good, (5) Feeling great |
| From 1 (Not at all) to 5 (Extremely), in the past 15 minutes, I was anxious, easily upset. |
| From 1 (Not at all) to 5 (Extremely), in the past 15 minutes, I was calm and emotionally stable. |

A full list of the over 80 different EMAs (mental health and non-mental health related) asked throughout the StudentLife study can be found on the StudentLife website [37].

aligned StudentLife variable, and similar to previous work, excluded data from any day of StudentLife data that did not contain at least 19 hours of collected data [3].

This resulted in 5 features for each of the following: activity duration on foot, still, and unknown, duration and number of conversations, location distance, phone unlock duration, and number of unique locations. In addition, we included features representing the daily sleep start, end, and duration (43 total). Call/text message data was included in the downloadable StudentLife data file. The StudentLife publication did not describe this data and raw file formats were inconsistent across participants. Thus, we excluded call/text log data [4]. A summary of the sensor data types across studies, and whether each data type was used in the analysis, with reasoning, is described in Table 4.

EMAs in both studies were not administered daily, and EMAs from the CrossCheck study were delivered and responded to more consistently (every 2–3 days) compared to StudentLife EMAs. Thus, similar to previous work predicting EMAs collected from the CrossCheck study, we calculated the mean of each behavioral feature across the three days up to and including an EMA response to align features and EMAs for prediction [3]. For example, if a participant responded to an EMA on day 6, the mean behavioral feature values from days 4–6 were used as model inputs to predict that EMA.

Data was occasionally missing for an individual, or our 19-hour coverage rule removed a day of data. We could fill data (e.g. interpolation) to mediate this issue, but filling may bias the data towards common values, making it difficult for models to identify feature variations indicative of mental health changes [5]. Similar to previous work, we created a 44th feature, describing the number of missing days of data within the averaged 3-day period [5,41]. For the StudentLife sleep features specifically, data was occasionally missing for all days within the 3-day period. In this case, we simply filled the 3-day average sleep features with the mean sleep feature value for that individual. Filling missing data in longitudinal behavioral data streams is an active area of research, and future work should clarify best practices [42]. All features are summarized in Fig 2.

## Model training and validation

We trained gradient boosting regression trees (GBRT) to predict self-reported EMA symptoms. We used GBRTs following prior research predicting mental health symptoms from mobile sensing data [2,3]. GBRTs sequentially train ensembles of shallow decision trees. Each added tree corrects mistakes from trained trees by upweighting incorrectly predicted samples.

**Table 4. The sensor data alignment between CrossCheck and StudentLife sensing data.**

| Data Element | Data Element Description | Data Element Used in Analysis? |
|---|---|---|
| Activity | The Android activity recognition API records information about whether a user is: on foot, walking, running, still, in vehicle, on bicycle, tilting, and unknown. | Yes: Specifically, the on foot, still, and unknown activity data. Walking and running values were zeroed-out in CrossCheck data, and thus we summed StudentLife walking and running variables to create an equivalent on foot variable. Bicycle, tilting, and in vehicle variables were not available in the StudentLife data. |
| Audio Amplitude | The ambient sound from a user's environment. | No: Not available in the StudentLife data. |
| Bluetooth | MAC addresses of surrounding bluetooth devices. | No: Not available in CrossCheck data. |
| Call/Text Logs | When a call/text occurred, and the call/text type. | No: Lack of StudentLife data documentation, and no prior use in previous StudentLife literature. |
| Conversation | Conversational episodes and duration. | Yes |
| Light | The mean and standard deviation of ambient light from a participant's environment (CrossCheck), or whether a user is in a dark room (StudentLife). | No: No mapping between StudentLife and CrossCheck variables. |
| Location | The distance a user traveled, as well as the number of unique locations visited. | Yes |
| Phone Charge | The duration a phone was charging for a significant amount of time. | No: Not available in CrossCheck data. |
| Phone Lock | The duration a phone was locked (StudentLife data) or unlocked (CrossCheck data) | Yes: Time between phone locks in the StudentLife data was used to estimate the unlock duration. |
| Sleep | On each day, the sleep duration, onset, and wake time were detected. | Yes: Not publicly available in the StudentLife data, but estimated from phone lock data [40]. |
| WiFi Location | WiFi scan logs detailing where an individual is located. | No: Not available in CrossCheck data. |

The first column describes the unioned data elements across the CrossCheck and StudentLife data, the second column describes the description of that element, and the third column whether the element was used in the analysis with reasoning. API: Application programming interface; MAC: Media access control.

Final predictions are obtained by adding predictions from trees in the order of training [3]. We varied hyperparameters including the learning rate (0.001, 0.01, 0.1, 1.0), number of trees trained (20, 100, 1000), and individual tree depth (3, 7, 10). All trees were trained using a Huber loss [3].

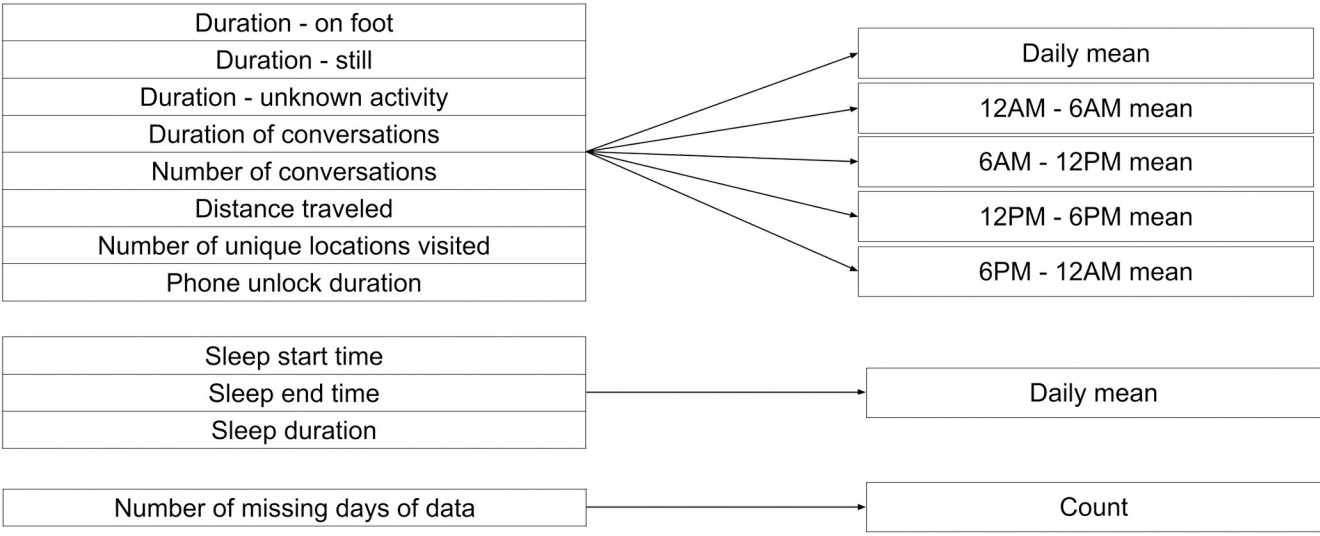

**Fig 2. Summary of the 44 features used for prediction.** Each data type on the left-hand side is summarized over a 3-day period for each epoch (e.g. 12AM - 6AM) using the aggregation technique (mean or count) described on the right-hand side. Aggregations were performed to align features with ecological momentary assessment (EMA) mental health symptom outcomes.

Leave-one-subject-out cross-validation (LOSO-CV) assessed model performance. LOSO-CV simulates the prediction error of applying models to participants unseen during model training [43]. We iterated through each participant, training models for each set of hyperparameters excluding that participant's data, and then applied trained models to predict the participant's EMAs. For each participant and set of hyperparameters, two models (Fig 1, step 1) were trained: (1) a *single-study* model using data exclusively from the study that the participant belonged to, and (2) a *combined* model using data across both (CrossCheck and StudentLife) studies. Participants were included as validation data if they had ≥30 EMA values collected. Without this threshold, it would be difficult to measure within-participant model prediction error [3].

## Oversampling to reduce class imbalance

Self-reported severe mental health symptoms are often under-represented in mobile sensing longitudinal studies, resulting in prediction models that underestimate symptom severity [2]. We used the synthetic minority oversampling technique (SMOTE) to augment each training dataset to balance EMA values prior to model training (Fig 1, step 2) [2]. SMOTE is a common oversampling technique that iterates through minority class data points, generating synthetic data points on the line between each minority data point and its k-nearest neighbors within the same class [36]. Similar to prior work, we set k = 5, standardized features (mean 0, standard deviation of 1) prior to SMOTE, and treated using/not using SMOTE as a hyperparameter [36].

## Personalizing models by aligning feature distributions

Mental health-mobile sensing relationships are heterogeneous across individuals, even within a single-study, and combining data across studies might exacerbate these heterogeneities [3,44,45]. We experimented with a local personalization procedure (Fig 1, step 3), motivated by previous work personalizing models with multimodal, longitudinal data streams [46]. For each held-out participant, we only included the k-nearest neighbors to that participant's mobile sensing behavioral features for model training, thus "personalizing" the training data based upon each participants' input behavioral feature distributions. k was a model hyperparameter, and we experimented with k = (5, 10, 50, 100, 500). Features were standardized, and nearest neighbors were identified using the Euclidean distance. Models with/without (using the entire training dataset) personalization were compared.

## Results

Machine learning results were analyzed using sensitivity analyses, where we conducted paired significance tests to analyze whether, within a specific hyperparameter combination, changing a single hyperparameter (e.g. combined versus single-study training data) significantly changed results. Sensitivity analyses were performed to understand performance changes independent of specific hyperparameters used, as hyperparameter choices can change conclusions drawn from optimal models alone [47].

## Aligned data overview

Table 5 summarizes the aligned data. CrossCheck participants consistently reported 10 EMAs every 2–3 days, resulting in 5,853 total EMAs collected. Applying the ≥30 EMA validation criteria, 5,665 EMAs were collected across 51 participants, median interquartile range (IQR) of 124 (80–141) responses per-participant. Only 3 EMAs—sleep quality, stress, and calmness—

**Table 5. Summary of the aligned training and validation data.**

| | CrossCheck | StudentLife (Sleep)[a] | StudentLife (Stress) |
|---|---|---|---|
| Total Number of Training Instances | 5,853 | 1,079 | 902 |
| Total Number of Validation Instances (Participants with $\geq$30 EMAs) | 5,665 | 597 | 307 |
| Number of Validation Participants | 51 | 15 | 9 |
| Instances across Validation Participants by Median (IQR) | 124 (80–141) | 38 (33–48) | 33 (32–36) |
| Sleep Validation EMA median (IQR)[b] | 2 (2–3) | 2 (2–3) | NA |
| Severe Sleep Validation Self-Reports (%)[c] | 1,240 (22) | 126 (21) | NA |
| Stress Validation EMA median (IQR) | 1 (0–1) | NA | 1 (1–2) |
| Severe Stress Validation Self-Reports (%) | 1,292 (23) | NA | 200 (43) |

NA: Not applicable.

a. Characteristics are listed separately for each ecological momentary assessment (EMA) predicted in the StudentLife population ("Sleep" and "Stress") as not all individuals who responded to sleep EMAs on a given day also responded to stress EMAs on the same day (and vice versa).

b. CrossCheck Sleep EMA values exist between 0 (low quality sleep) to 3 (high quality sleep), and stress EMA values exist between 0 (low stress) and 3 (high stress). StudentLife EMA values were scaled between 0 and 3 to match the CrossCheck data.

c. Sleep and stress responses were considered "severe" if they fell into either of the two EMA categories indicating poorer sleep (0–1) or higher stress (2–3) respectively.

had similar question-response structure across studies, and were considered as candidate EMAs for cross-dataset prediction. 1,079 sleep and 902 stress StudentLife EMAs were collected. 15 StudentLife participants self-reported $\geq$30 sleep EMAs (597 total, median [IQR] 38 [33–48]), and 9 $\geq$30 stress EMAs (307 total responses, median [IQR] 33 [32–36]). The calm EMA had <30 responses collected across all StudentLife participants, and was not used.

## Aligned sensor and EMA distribution differences

All mobile sensing features were non-normally distributed (omnibus test for normality p<0.001) [48]. We calculated Wendt's formulation of the rank-biserial correlation (RBC[−1, 1]) to quantify the magnitude of the feature distribution differences across datasets [49]. All features except for the distance traveled were significantly different (Mann-Whitney U test, two-sided, or Chi-square test of independence, $\alpha$ = 0.05) between datasets (see Fig 3). Outliers may be an important indicator of mental health changes, and were not excluded [5].

EMA responses were treated as continuous variables and normalized to a range from 0–3 within each dataset. EMA values were non-normally distributed (omnibus test for normality p<0.001) [48]. Sleep (U = 1,807,220, p = 0.003, RBC = -0.07) and stress (U = 827,394, p<0.001, RBC = 0.37) EMA distributions were significantly different (Mann-Whitney U test, two-sided, $\alpha$ = 0.05) across datasets (see Fig 4). Severe sleep/stress symptoms (scores 2–3) were self-reported less frequently in both the CrossCheck (22/23%) and StudentLife (21/43%) datasets.

## Combined training data more likely to be predictive than single-study data

Table 6 shows, out of the 432 hyperparameter combinations tested, within each model, training with combined data significantly ($\alpha$ = 0.05) outperformed baseline mean prediction models more frequently compared to models trained with single-study data. Across models, we tested the alternative hypothesis that the combined data significantly decreased the LOSO-CV mean absolute error (MAE) compared to single-study data ($\Delta MAE = MAE_{Single} - MAE_{Combined}$). To equalize the influence of each subject, for each hyperparameter combination, we first calculated the MAE for each subject, and then averaged MAEs across subjects. Model MAE distributions were non-normal (Shapiro-Wilk p<0.05), and we performed a non-parametric Wilcoxon signed-rank test (one-sided).

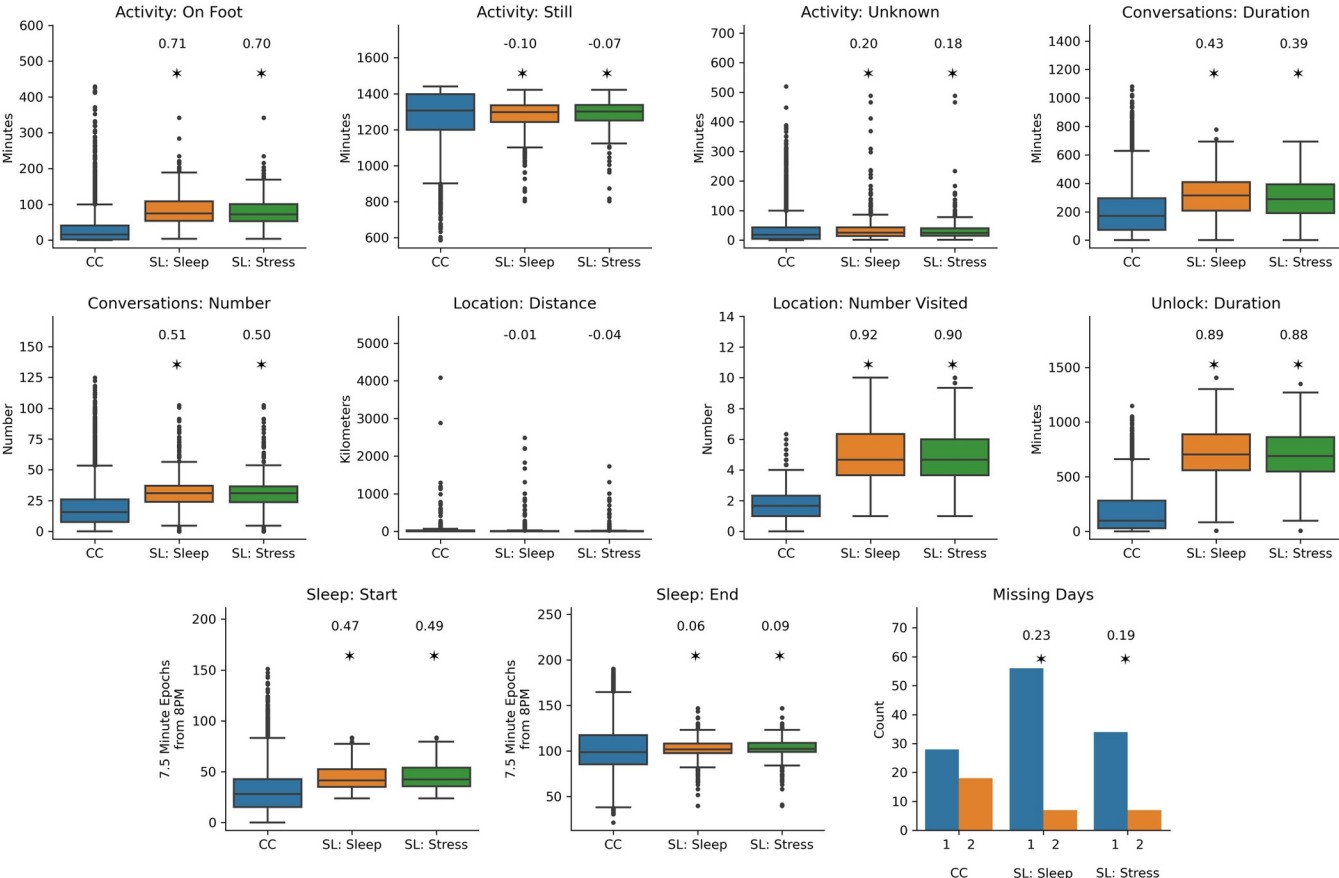

**Fig 3. Example feature distribution differences across datasets.** Assessing feature distributional differences across the CrossCheck (CC), StudentLife sleep EMA (SL: Sleep), and stress EMA (SL: Stress) validation data for an example 11 features across data types. Each subfigure shows a boxplot of the feature distribution within each specific dataset. The centerline of the boxplot is the median, the box edges the interquartile range (IQR), and the fences on the boxplot are values 1.5 x the IQR. The "Missing Days" distribution is a histogram, describing counts across participants. A "*" is listed above each of the StudentLife datasets if the distribution differed significantly (Mann-Whitney U test, two-sided, or Chi-square test of independence, α = 0.05) from CrossCheck. The numbers above the "*" are the rank-biserial correlation (RBC) or Cramer's V, which shows the magnitude of these differences. EMA: Ecological momentary assessment.

Fig 5 shows the sorted ΔMAE distribution percentiles across hyperparameters for each EMA and study, with p-values and matched-pairs RBC[−1, 1], quantifying the proportion of summed favorable minus unfavorable ranks [49]. Using combined data significantly (α = 0.05) improved model performance for predicting sleep within CrossCheck (W = 53,200, p = 0.007, RBC = 0.14) and StudentLife (W = 63,089, p<0.001, RBC = 0.35), and stress within Cross-Check (W = 55,373, p<0.001, RBC = 0.18), but not StudentLife. Fig 5 also shows the MAE of combined versus single-study models across percentiles, compared to a baseline MAE of pre-dicting the training data mean value. Minimum MAE values across different training datasets (single-study versus combined) were similar.

## Combined data improves model performance if feature distribution alignment increases

We experimented with quantifying when models improved using combined versus single-study data. We calculated the Proxy-A distance (PAD) between each LOSO-CV held-out study participant and each model training dataset used. The PAD is 2(1−2ε), where ε is the

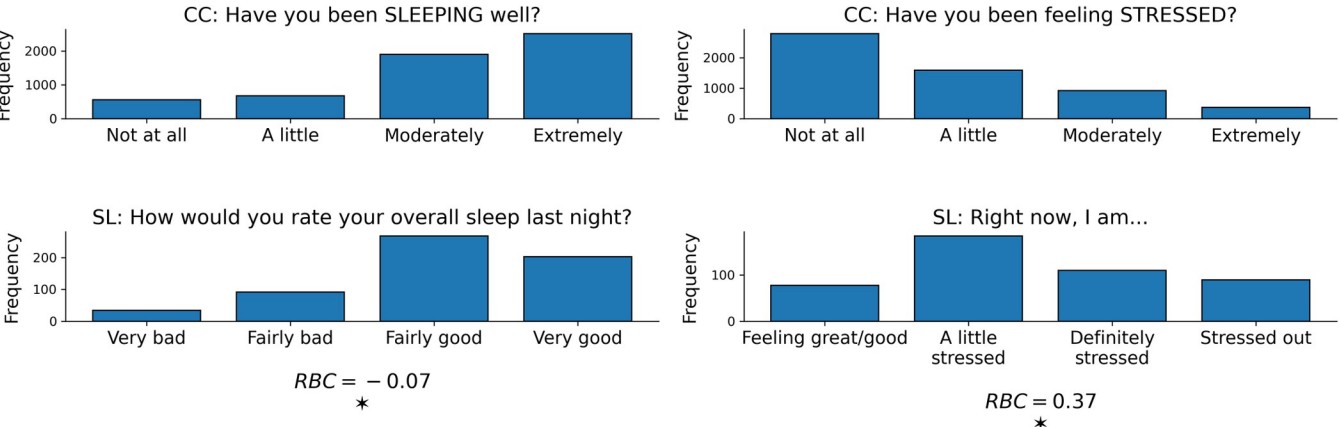

**Fig 4. Outcome distribution differences across datasets.** Sleep (left column) and stress (right column) ecological momentary assessment (EMA) validation distributions for CrossCheck (CC, top row) and StudentLife (SL, bottom row) data. The height of each bar represents the EMA response, where the specific response is listed on the x-axis under that bar. On the bottom, a "*" indicates whether there were significant (Mann-Whitney U test, two-sided, α = 0.05) differences between CrossCheck and StudentLife EMA distributions, with rank-biserial correlation (RBC) values listing the magnitude of these differences.

generalization MAE for a linear support vector machine (SVM) trained to distinguish between training and validation data. As the PAD decreases, the SVM has greater difficulty distinguishing datasets, implying the data distributions were increasingly similar [52]. We used both generalized estimating equations (GEE) and linear mixed-effect models (LMM) to estimate the association between $\Delta PAD = PAD_{Single}\text{-}PAD_{Combined}$ and $\Delta MAE = MAE_{Single}\text{-}MAE_{Combined}$ within subjects. GEE is a clustered linear regression model, often used instead of LMM because it places less assumptions on the data-generating distribution, but GEE may have a larger Type 1 error than LMM, resulting in falsely significant associations [50]. Both GEE and LMM results showed the same significant ($p_{GEE}$ = 0.004, $p_{LMM}$ = 0.007) $\Delta MAE$ (95% CI) increase of 0.07 (0.02 to 0.12) per unit increase in $\Delta PAD$ (see Table 7).

**Table 6. Sensitivity analysis of predictive models using different training datasets.**

| EMA | LOSO-CV Data | $\Delta MAE_{BC}$ | $\Delta MAE_{BS}$ | $\Delta MAE_{SC}$ | $\Delta MAE_{CS}$ | $\Delta MAE_{BC\cap SC}$ | $\Delta MAE_{BS\cap CS}$ |
|---|---|---|---|---|---|---|---|
| Without Benjamini–Hochberg correction (α = 0.05)* | | | | | | | |
| Sleep | CrossCheck | 67 (16) | 24 (6) | 19 (4) | 18 (4) | 2 (0) | 0 (0) |
| Sleep | StudentLife | 35 (8) | 19 (4) | 46 (11) | 44 (10) | 1 (0) | 0 (0) |
| Stress | CrossCheck | 86 (20) | 28 (6) | 21 (5) | 18 (4) | 0 (0) | 0 (0) |
| Stress | StudentLife | 16 (4) | 9 (2) | 4 (1) | 7 (2) | 0 (0) | 0 (0) |
| With Benjamini–Hochberg correction (FDR = 25%) | | | | | | | |
| Sleep | CrossCheck | 29 (7) | 0 (0) | 11 (3) | 2 (0) | 1 (0) | 0 (0) |
| Sleep | StudentLife | 14 (3) | 0 (0) | 29 (7) | 0 (0) | 0 (0) | 0 (0) |
| Stress | CrossCheck | 60 (14) | 0 (0) | 6 (1) | 0 (0) | 0 (0) | 0 (0) |
| Stress | StudentLife | 2 (0) | 0 (0) | 0 (0) | 0 (0) | 0 (0) | 0 (0) |

Values are listed as "number (%) of significant models", and are shown without and with using a Benjamini–Hochberg correction to correct for a false discovery rate (FDR) of 25% [50]. 432 total hyperparameter combinations were tested per training dataset and ecological momentary assessment (EMA). Training data could either be: "B" a baseline model predicting the mean training data EMA value, "C" the combined data, or "S" single-study data. Alternative hypotheses were tested following $\Delta MAE_{ij} = MAE_i\text{-}MAE_j > 0$. The last two columns show models where the intersection was significant: $\Delta MAE_{ij \cap xy} = \Delta MAE_{ij}$ significant and $\Delta MAE_{xy}$ significant. All significance tests were performed using a Rosner test, a non-parametric Wilcoxon signed-rank test (one-sided) that accounts for within-cluster (participant) rank variation [51]. EMA: Ecological momentary assessment; LOSO-CV: Leave-one-subject-out cross-validation; MAE: Mean absolute error.

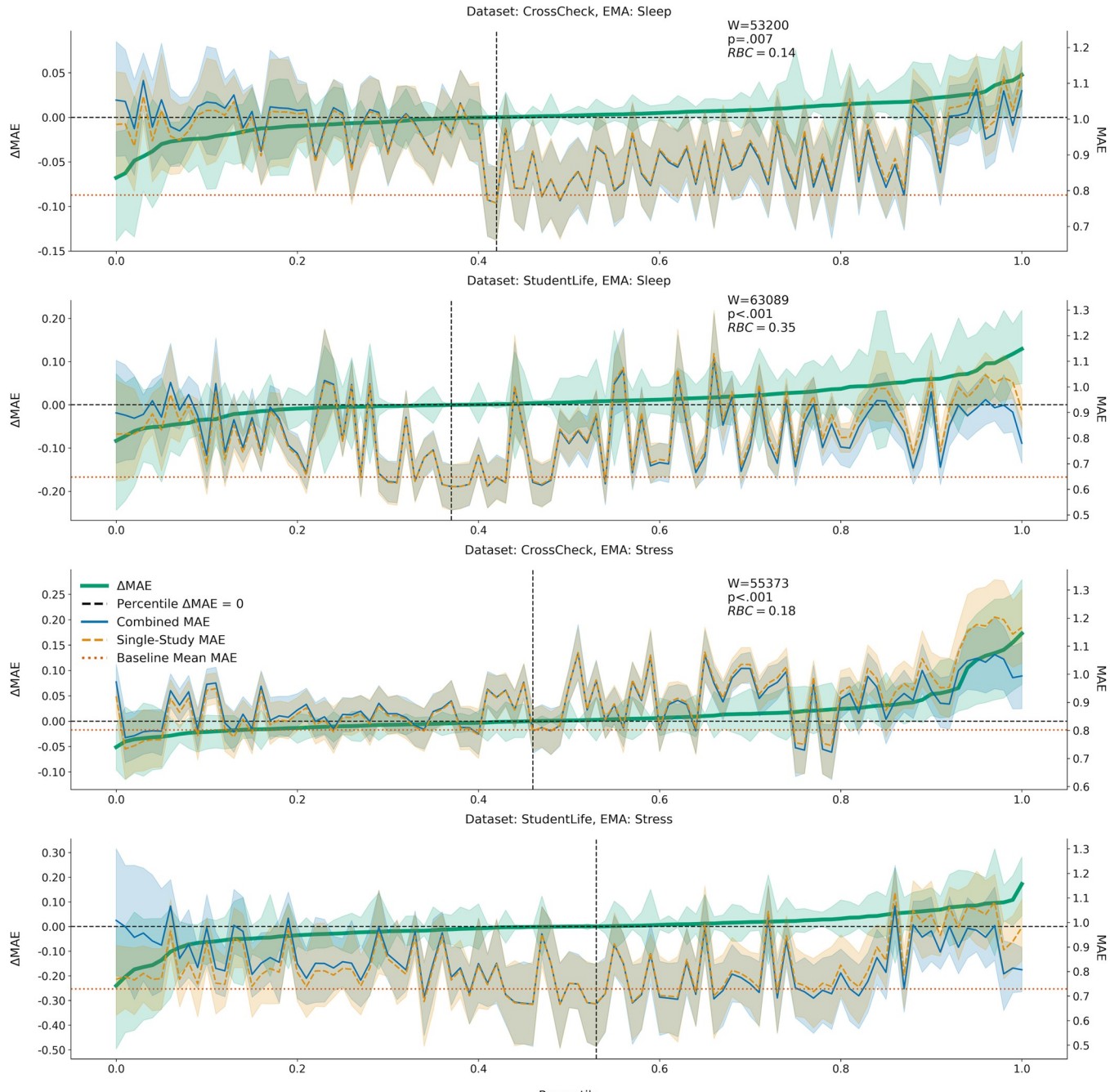

**Fig 5. Sensitivity analysis reveals the combined versus single-study data is more likely to be predictive.** The left y-axis describes the $\Delta MAE = MAE_{Single}$-$MAE_{Combined}$ against the sorted distribution percentiles (x-axis). The thick green solid line represents the $\Delta MAE$ percentiles, and the dashed black intersection lines show the percentile value (x-axis) where $\Delta MAE = 0$. The right y-axis describes the actual MAE for the combined (blue solid line), and single-study (dashed orange line) data at each percentile. The baseline MAE, or error for a model predicting the average of the training data, is described by the dotted horizontal red line. Wilcoxon signed-rank test (one-sided) statistics (W), p-values, and rank-biserial correlations (RBCs) are included for models where across hyperparameters, using combined data significantly ($\alpha = 0.05$) outperformed using single-study data (one-sided test). Shaded areas represent 95% confidence intervals around the mean. EMA: Ecological momentary assessment.

**Table 7. Uncovering the association between distributional distance (ΔPAD) and model performance (ΔMAE).**

|  | $\beta_{GEE}$ (95% CI) | $p_{GEE}$ | $\beta_{LMM}$ (95% CI) | $p_{LMM}$ |
|---|---|---|---|---|
| **ΔPAD** | 0.07 (0.02 to 0.12) | 0.004 | 0.07 (0.02 to 0.12) | 0.007 |
| **EMA (Sleep/Stress)** | 0.01 (-0.01 to 0.02) | 0.475 | 0.01 (0.00 to 0.01) | 0.222 |
| **neighbors = 10** | 0.01 (0.00 to 0.02) | 0.162 | 0.01 (-0.01 to 0.02) | 0.191 |
| **neighbors = 50** | 0.01 (-0.01 to 0.02) | 0.502 | 0.01 (-0.01 to 0.02) | 0.430 |
| **neighbors = 100** | 0.01 (-0.01 to 0.03) | 0.427 | 0.01 (-0.01 to 0.02) | 0.331 |
| **neighbors = 500** | 0.01 (-0.01 to 0.03) | 0.372 | 0.01 (-0.01 to 0.02) | 0.293 |
| **neighbors = All** | 0.00 (-0.01 to 0.02) | 0.535 | 0.00 (-0.01 to 0.02) | 0.562 |
| **constant term** | 0.00 (-0.02 to 0.01) | 0.793 | 0.00 (-0.01 to 0.01) | 0.762 |

Generalized estimating equations (GEE) and linear mixed-effect model (LMM) results estimating the association between the change in Proxy-A distance ($\Delta PAD = PAD_{Single}-PAD_{Combined}$) and change in mean absolute error ($\Delta MAE = MAE_{Single}-MAE_{Combined}$). Coefficients (β) are displayed by the mean (95% Confidence Interval) with significance levels (p). We controlled for the ecological momentary assessment (EMA) predicted (sleep versus stress), and the number of neighbors used for model personalization.

## Personalization not guaranteed to improve model performance

We explored if local personalization to align training and held-out feature distributions improved model performance. Fig 6A shows that constraining the number of neighbors resulted in more-aligned training and validation participant distributions (decreased Proxy-A distance). Despite increased alignment, including more neighbors generally decreased the model MAE (Fig 6B) across symptoms and datasets. Personalization with 5 or 10 neighbors outperformed CrossCheck stress models trained with the entire dataset.

## Oversampling imbalanced EMA values increases sensitivity, reduces specificity

We used the synthetic minority oversampling technique (SMOTE) to oversample minority EMA values and equalize value representation in each training dataset. Fig 7 shows the MAE significantly increased using SMOTE across EMAs and datasets. Regression models often output the lowest MAE by predicting the training data mean [2]. We analyzed if SMOTE improved LOSO-CV performance by transforming the regressions into a binary classification problem, coding the two most severe symptom responses for each EMA with a "1", and the other responses with a "0". Fig 7 compares the sensitivity, specificity, and positive predictive value (PPV) of using/not using SMOTE. Metric distributions across hyperparameters were non-normally distributed (Shapiro-Wilk $p<0.05$). A paired Wilcoxon signed-rank test (one-sided) found that the sensitivity was significantly greater (α = 0.05) using SMOTE across all EMAs and datasets. SMOTE significantly increased the PPV for predicting stress, and marginally (α = 0.10) increased the PPV for predicting sleep across datasets. Using SMOTE significantly decreased the specificity across all EMAs and datasets.

## Discussion

We present a first-of-a-kind analysis combining data across longitudinal mobile sensing studies to predict mental health symptoms. We aligned calculated behavioral features and symptom self-reports between datasets, and conducted a sensitivity analysis to quantify the expected gain in model performance across hyperparameters. Prior studies calculated a variety of sensor features summarizing different types of information (e.g. summary statistics, circadian rhythms) [3,5,15,44]. The CrossCheck public data included calculated daily summary

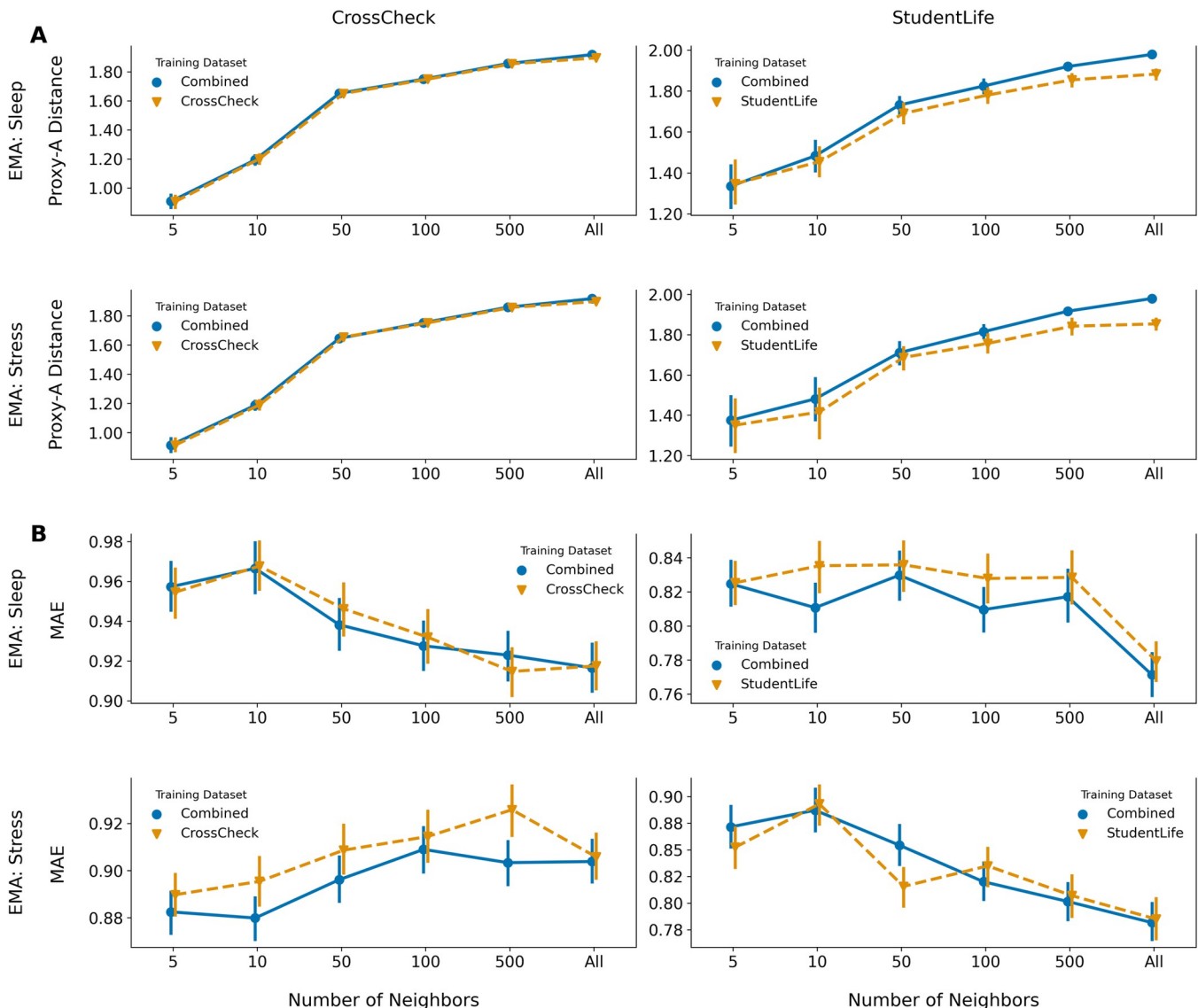

**Fig 6. Personalization increases training and held-out data alignment, but is not guaranteed to improve prediction performance.** (A) Effects of personalization by changing the number of neighbors (x-axis) used for model training on the feature distribution alignment between training and leave-one-subject-out cross-validation (LOSO-CV) participants (Proxy-A distance, y-axis). (B) Effects of changing the number of neighbors (x-axis) during model training on the model mean absolute error (MAE, y-axis). On all plots, each point is the mean Proxy-A distance (A) or MAE (B) across hyperparameters, and error bars are 95% confidence intervals around the mean. Each plot is split by the training data used (combined versus single-study), and plots are specific to the LOSO-CV result for a study (CrossCheck/StudentLife) and EMA (Sleep/Stress).

features, and StudentLife close-to-raw sensor data, which allowed us to calculate corresponding CrossCheck features from StudentLife data. While publicly sharing close-to-raw sensor data enables data alignment, it also raises privacy concerns. For example, the StudentLife data contained GPS location, which could be paired with publicly available geotagged identifying information for within dataset re-identification [53]. Data sharing may enable future work to continue to assess model generalizability, but governance suggesting data de-identification standards and access controls is needed to ensure appropriate data reuse [54].

Outcome symptom measures were less easy to align across studies. This is not surprising; clinical studies intentionally measure symptoms of a specific serious mental illness (SMI),

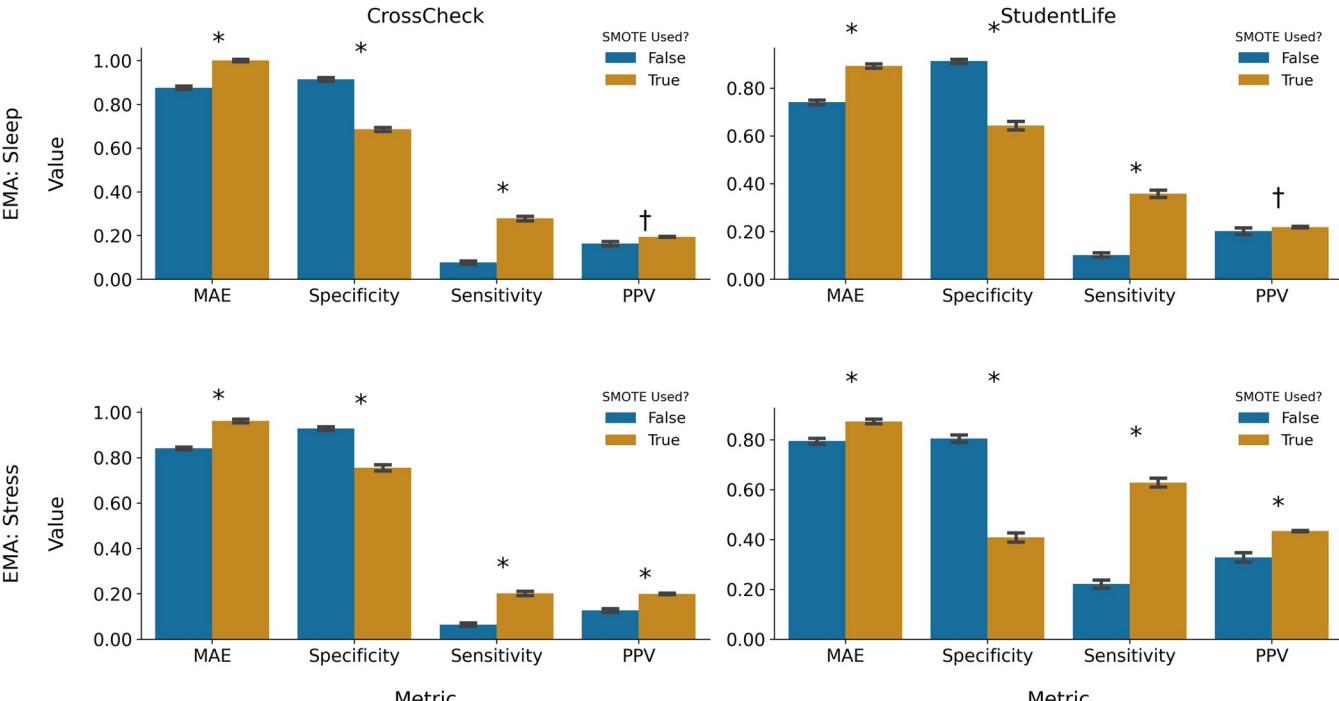

**Fig 7. SMOTE increases sensitivity, positive predictive value, but reduces specificity and increases mean absolute error.** SMOTE (see legends) oversampled under-represented ecological momentary assessment (EMA) values. The height of each bar is the mean value of the metric described on the x-axis across hyperparameters. Error bars are 95% confidence intervals around the mean. Plots are specific to the leave-one-subject-out cross-validation (LOSO-CV) result for a study (CrossCheck/StudentLife) and ecological momentary assessment (EMA) (Sleep/Stress). The specificity, sensitivity, and positive predictive value (PPV) were calculated by transforming regression results into a classification problem by labeling the two most severe symptom classes in each EMA with a "1" and other symptoms as "0". Otherwise, the plots analyzed the regression mean absolute error (MAE). "*" indicates p<0.05, and "†" indicates p<0.10, for a Wilcoxon signed-rank test (one-sided) exploring differences using SMOTE/not using SMOTE across hyperparameter combinations.

while non-clinical studies collect measures on more prevalent symptoms across the general population (e.g. depression, stress) [3,5,17,20]. That being said, symptoms of depression are symptoms of SMIs, including schizophrenia [55]. In this analysis, alignment of shared symptoms across studies was difficult, as each study used a different EMA symptom questionnaire battery [3,4]. The international mental health research community has encouraged agencies to require standardized outcome measures (e.g. PHQ-9) within funded research, but suggested measures, due to their length, may be arduous to frequently self-report or are intended to assess symptoms at longer time-scales, misaligned with the opportunity of mobile sensing for frequent assessment [56]. Developing a standardized battery of in-the-moment symptom measures for continuous remote symptom assessment studies would advance research on model generalizability.

Sensitivity analyses revealed that the combined data were more likely to improve EMA prediction (Fig 5) compared to single-study data, and were more likely to be predictive (Table 6) over the baseline models. Machine learning models are costly to train, specifically with extensive hyperparameter tuning [57]. Our results showed that combining mobile sensing datasets may offer a more efficient pathway from model building to deployment, defining efficiency as the number of hyperparameters searched to identify a predictive model. Despite this efficiency gain, optimal MAE values were similar using single-study versus combined data. Thus, as other research shows, we cannot naively expect more data to optimize performance [58]. Future mobile sensing research could experiment with other data alignment methods to understand if/when performance gains may occur [59].

Reducing the PAD, by using combined versus single-study data for model training, significantly reduced the model MAE, implying that model performance improved when the combined data had greater alignment with validation data compared to single-study data (Table 7). The transfer learning subfield of *domain adaptation* offers a variety of approaches to continue this line of research by aligning data collected from heterogeneous sources for the same prediction task [35,46]. Domain adaptation approaches could be used for cross-dataset prediction to align feature distributions across participants, or datasets. Another transfer learning approach often used in remote mental health symptom assessment literature, called *multitask learning*, treats prediction tasks within heterogeneous study datasets as separate-but-related tasks [60]. The prediction of each study participant's symptoms, or cluster of participants that share behavior-mental health relationships, is defined as a separate prediction task [41,44,45]. Participants unseen during model training must then be matched to a cluster for prediction, which is difficult when minimal to no mobile sensing or symptom data has been collected for that participant. Future work should focus on how domain adaptation and/or multitask learning can be leveraged for accurate modeling in datasets with increased sources (e.g. population, device) of heterogeneity, working to minimize the anticipated data collection burden on participants.

Our results offer a clue to how transfer learning may be applied to improve model performance. Specifically, we found that personalization by aligning the behavioral feature space alone (Fig 6A) did not always improve model performance (Fig 6B). The lack of performance gain despite better feature alignment highlights that behavior-mental health relationships across individuals may vary. This corresponds with clinical literature highlighting the heterogeneous presentation of mental health symptoms across individuals, even within the same disorder [26,61]. Understanding how symptom heterogeneity manifests within behavioral mobile sensing features may be essential for more-accurate prediction. Our results point future work towards modeling approaches that align both features and outcome symptoms when designing prediction tasks.

We used SMOTE to oversample minority EMA values representing more severe mental health symptoms. Prior work shows that prediction models underpredict severe mental health symptoms [2]. From a classification perspective, models predicting extreme symptom changes often result in low sensitivity, but high specificity [5,13]. Similarly, we found SMOTE improved model sensitivity and PPV, but reduced specificity (Fig 7). While these results may be obvious—biasing the training data towards a specific outcome likely improves prediction of the oversampled outcome—to the best of our knowledge, the results and implications of using oversampling techniques for longitudinal mental health symptom prediction have not been discussed in the literature, and oversampling may be useful despite the specificity decrease. Alert systems, triggering interventions in response to predicted symptom changes, could account for higher false positives through low friction responses, for example, a patient reachout by a care manager [5]. Lower specificity is less problematic than lower sensitivity, the latter resulting in undetected patients in need of care. Through this frame, oversampling, and data augmentation more broadly, could be beneficial [29].

This research implies that previous de-identified mobile sensing study data can potentially be deployed to predict symptoms across multiple populations. In-practice, clinicians may be able to reuse models pretrained on external populations to predict symptoms within their own patients, though future research should explore the amount of within-population data needed for accurate prediction. Reuse is particularly useful when deploying models in populations typically underrepresented in mobile sensing studies, including elderly or less-affluent communities [27]. This research does not imply that combining heterogeneous data improves model performance compared to training a machine learning model on a larger homogenous sample.

In fact, this research implies the opposite. The decrease in PAD after combining datasets implies that the larger combined training data used in this paper was more representative of out-of-sample participants. Researchers should continue to test models across more diverse datasets to understand when combining data improves or degrades model performance. Model performance can degrade if the combined populations are too dissimilar—known as negative transfer in the machine learning literature [62].

This study had limitations. First, demographic data was not reported in either public dataset, and we could not assess prediction equity across demographic subpopulations. Both studies were small, and individual study populations were relatively homogenous. A number of potentially useful data types (audio amplitude, bluetooth, call/text logs, light, phone charge, WiFi location) were misaligned across datasets, and not included as features. Future work can explore more complex modeling techniques to include both aligned and misaligned features across datasets for prediction. Finally, the StudentLife and CrossCheck studies were conducted by a similar research collaboration, implying that the studies' design may be similar. As mobile sensing studies across different research groups become publicly available, more diverse datasets can be combined to further assess generalizability.

In conclusion, we found that machine learning models trained across longitudinal mobile sensing study datasets may generalize, and provide a more efficient method to build predictive models. By assessing generalizability, we move the field closer to deploying remote, longitudinal mental health symptom assessment systems.

## Acknowledgments

Thank you to the researchers who collected the StudentLife and CrossCheck datasets for publicly releasing de-identified data.

## Author Contributions

**Conceptualization:** Daniel A. Adler, Fei Wang, Tanzeem Choudhury.

**Formal analysis:** Daniel A. Adler.

**Funding acquisition:** Daniel A. Adler, Tanzeem Choudhury.

**Investigation:** Daniel A. Adler.

**Methodology:** Daniel A. Adler, Fei Wang.

**Resources:** Tanzeem Choudhury.

**Software:** Daniel A. Adler.

**Supervision:** Fei Wang, David C. Mohr, Tanzeem Choudhury.

**Validation:** Daniel A. Adler.

**Visualization:** Daniel A. Adler.

**Writing – original draft:** Daniel A. Adler.

**Writing – review & editing:** Daniel A. Adler, Fei Wang, David C. Mohr, Tanzeem Choudhury.

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
