## [Decision Letter · Decision Letter 0]

17 Feb 2022

PONE-D-22-01484Machine learning for passive mental health symptom prediction: generalization across different longitudinal mobile sensing studiesPLOS ONE

Dear Dr. Adler,

Thank you for submitting your manuscript to PLOS ONE. After careful consideration, we feel that it has merit but does not fully meet PLOS ONE’s publication criteria as it currently stands. Therefore, we invite you to submit a revised version of the manuscript that addresses the points raised during the review process.

We look forward to receiving your revised manuscript.

Kind regards,

Chi-Hua Chen, Ph.D.

Academic Editor

PLOS ONE

Journal Requirements:

"DA is co-employed by United Health Group, outside of the submitted work. TC is a co-founder and equity holder of HealthRhythms, Inc., is co-employed by United Health Group, and has received grants from Click Therapeutics related to digital therapeutics, outside of the submitted work. DA and TC hold pending patent applications related to the cited literature. DCM has accepted honoraria and consulting fees from Apple, Inc., Otsuka Pharmaceuticals, Pear Therapeutics, and the One Mind Foundation, royalties from Oxford Press, and has an ownership interest in Adaptive Health, Inc. FW declares no competing interests."

Reviewers' comments:

Reviewer's Responses to Questions

**Comments to the Author**

1. Is the manuscript technically sound, and do the data support the conclusions?

Reviewer #1: Partly

Reviewer #2: Yes

2. Has the statistical analysis been performed appropriately and rigorously? 

Reviewer #1: Yes

Reviewer #2: Yes

3. Have the authors made all data underlying the findings in their manuscript fully available?

Reviewer #1: Yes

Reviewer #2: Yes

4. Is the manuscript presented in an intelligible fashion and written in standard English?

Reviewer #1: Yes

Reviewer #2: Yes

5. Review Comments to the Author

Reviewer #1: Mental illnesses are among the crucial healthcare concerns today. Conventional approaches for the detection of mental problems by clinicians and psychologists suffer from the limitations of the accessible information about the individual's thoughts and experiences. In the recent years, mobile sensing technologies have made a good progress in collecting different kinds of individual data related to their daily life activities. This paper presents a study on using machine learning to predict mental health symptoms from mobile sensing data. The main aim of this study is to assess the generalizability of the machine learning model to predict the symptoms across different mental problems and related data. Two publicly available datasets are considered for their experimentation with diferent approaches. The study finds that the model is more effective by fusing multiple heterogeneous datasets and oversampling less-represented severe symptoms. The conducted research is very important and timely. The presented ideas are interesting. However, I have some concerns.

- While it is clearly said that both the datasets were collected in other previous studies, I am little confused with pages 6-12. Are these pages part of the current proposed study or part of previous studies, just explained again by the authors?

- The sizes of the datasets are really small. In such case, whether the datasets are sufficiently representative for the different mental health symptoms, is not clear. What are the different symptoms reported in the datasets? How many instances of them are reported? How were the feature values set for those features that did not align accross the heterogeneous datasets? It would be interesting to look at the number of instances and size of their feature vectors used for training and testing the model.

- A claim made in the paper is that the model performs better by combining multiple heterogenous datasets. There are two possibilities for the better results obtained in the experiments. First, the authors are correct. In this case, two small heterogeneous datasets combined into one small (still small, but larger than the individual datasets) should perform better than one large homogeneous (single) dataset. Second, it could be just because of the larger dataset size of the combination of two datasets. So the combined data is just having more samples and thus more representative. It is a well established fact that machine learning models will perform better with more representative samples. From the available datasets and results it is difficult to conclude, which one of the two possibilities is true. To my understanding, I think the second possiblity is more likely to be true. The authors may want to look into it.

- Some studies use machine/deep learning on large-scale social media data along with other data types to predict mental health issues. The authors should refer some literature on machine learning research for mental health in the paper.

[1] Su, C., Xu, Z., Pathak, J. et al. Deep learning in mental health outcome research: a scoping review. Transl Psychiatry 10, 116 (2020)

[2] S. Ghosh and T. Anwar, Depression Intensity Estimation via Social Media: A Deep Learning Approach, in IEEE Transactions on Computational Social Systems, vol. 8, no. 6, pp. 1465-1474, Dec. 2021

Reviewer #2: This article offers a preliminary investigation to determine if models built using combined longitudinal study data to predict mental health symptoms generalise across current publically accessible data. The topic is up-to-date and of major interest to the majority of the journal audience. The paper is well organized and easy to follow. However, the authors need to provide a few details and to overcome a few shortcomings, which can make the paper better. The required shortcomings are given below.

1. I am unable to verify whether/how the authors plan to make the created data available. The dataset, if not made available for the research community, will reduce the contribution of this work significantly.

2. Figures in this paper have low quality.

3. The overall language quality could be improved. The structure of some sentences makes the reading really hard.

4. Some related studies could be discussed.

a. Ghosh, Soumitra, Asif Ekbal, and Pushpak Bhattacharyya. "What Does Your Bio Say? Inferring Twitter Users' Depression Status From Multimodal Profile Information Using Deep Learning." IEEE Transactions on Computational Social Systems (2021).

b. Ghosh, Soumitra, Asif Ekbal, and Pushpak Bhattacharyya. "Cease, a corpus of emotion annotated suicide notes in english." Proceedings of The 12th Language Resources and Evaluation Conference. 2020.

c. Ghosh, Soumitra, Asif Ekbal, and Pushpak Bhattacharyya. "A multitask framework to detect depression, sentiment and multi-label emotion from suicide notes." Cognitive Computation 14.1 (2022): 110-129.

5. Some more references in 2020 and 2021 should be considered.

6. PLOS authors have the option to publish the peer review history of their article (what does this mean?). If published, this will include your full peer review and any attached files.

Reviewer #1: No

Reviewer #2: No

---

## [Author Response · Author response to Decision Letter 0]

7 Mar 2022

Dear Dr. Chi-Hua Chen and the editors of PLOS ONE,

Thank you for providing the opportunity to revise our manuscript for the PLOS ONE “Remote Assessment Call for Papers”. All point-by-point responses below begin with the capitalized “RESPONSE” text following the corresponding journal/reviewer comment. 

Daniel A. Adler

On behalf of coauthors: Fei Wang, David C. Mohr, and Tanzeem Choudhury

Journal Requirements:

COMMENT J1. Please ensure that your manuscript meets PLOS ONE's style requirements, including those for file naming. The PLOS ONE style templates can be found at 

RESPONSE J1: We have updated our manuscript to follow the style requirements. Please reach out if there are any remaining styling corrections.

COMMENT J2. Thank you for stating the following in the Competing Interests section: 

"DA is co-employed by United Health Group, outside of the submitted work. TC is a co-founder and equity holder of HealthRhythms, Inc., is co-employed by United Health Group, and has received grants from Click Therapeutics related to digital therapeutics, outside of the submitted work. DA and TC hold pending patent applications related to the cited literature. DCM has accepted honoraria and consulting fees from Apple, Inc., Otsuka Pharmaceuticals, Pear Therapeutics, and the One Mind Foundation, royalties from Oxford Press, and has an ownership interest in Adaptive Health, Inc. FW declares no competing interests."

RESPONSE J2: We confirm that our Competing Interests statement does not alter our adherence to PLOS ONE policies on sharing data and materials. Our revised statement is below.

“DA is co-employed by United Health Group, outside of the submitted work. TC is a co-founder and equity holder of HealthRhythms, Inc., is co-employed by United Health Group, and has received grants from Click Therapeutics related to digital therapeutics, outside of the submitted work. DA and TC hold pending patent applications related to the cited literature. DCM has accepted honoraria and consulting fees from Apple, Inc., Otsuka Pharmaceuticals, Pear Therapeutics, and the One Mind Foundation, royalties from Oxford Press, and has an ownership interest in Adaptive Health, Inc. FW declares no competing interests. This does not alter our adherence to PLOS ONE policies on sharing data and materials.”

Reviewers' comments and responses:

Reviewer #1: Mental illnesses are among the crucial healthcare concerns today. Conventional approaches for the detection of mental problems by clinicians and psychologists suffer from the limitations of the accessible information about the individual's thoughts and experiences. In the recent years, mobile sensing technologies have made a good progress in collecting different kinds of individual data related to their daily life activities. This paper presents a study on using machine learning to predict mental health symptoms from mobile sensing data. The main aim of this study is to assess the generalizability of the machine learning model to predict the symptoms across different mental problems and related data. Two publicly available datasets are considered for their experimentation with diferent approaches. The study finds that the model is more effective by fusing multiple heterogeneous datasets and oversampling less-represented severe symptoms. The conducted research is very important and timely. The presented ideas are interesting. However, I have some concerns.

COMMENT 1A. While it is clearly said that both the datasets were collected in other previous studies, I am little confused with pages 6-12. Are these pages part of the current proposed study or part of previous studies, just explained again by the authors?

RESPONSE 1A: We have clarified in the manuscript the sections describing the previous studies, versus new analysis. See the beginning of the methods: 

“In this section, we first summarize the StudentLife and CrossCheck studies and data, which are the two longitudinal mobile sensing datasets analyzed in this work. Data collection was not completed in this study, and all analyses included in this study were completed on de-identified publicly released versions of the datasets, downloaded from [37,38]. Please see [3,4] for further details on data collection. We then describe the specific analyses used in this work to explore if models trained using combined (CrossCheck and StudentLife) longitudinal study data to predict mental health symptoms generalize. Specifically, we describe methods used to align collected sensor data and outcome measures across the two datasets, train and validate machine learning models, oversample minority outcomes to reduce class imbalance, and personalize models by aligning behavioral feature distributions.”

To provide a more detailed explanation for the reviewer, we provided the following subsections to summarize the relevant pieces of the previous studies for this work, including the study populations, study duration, sensing data, and symptom outcome measures collected in those studies. Specifically, the summary is in the following subsections and Table (original manuscript pages 5-10):

CrossCheck study and dataset

CrossCheck sensing data

CrossCheck EMA data

StudentLife study and dataset

StudentLife sensing data

StudentLife EMA data

Table 1

Within these sections, we also cited references to the original publications detailing these studies. It is common practice to re-summarize mobile sensing data collected in prior studies if the paper provides secondary analysis of the collected data. Please see the following published secondary analysis papers which do the same:

[1] Morshed MB, Saha K, Li R, D’Mello SK, De Choudhury M, Abowd GD, et al. Prediction of Mood Instability with Passive Sensing. Proc ACM Interact Mob Wearable Ubiquitous Technol. 2019;3: 1–21. doi:10.1145/3351233

[2] Adler DA, Ben-Zeev D, Tseng VW-S, Kane JM, Brian R, Campbell AT, et al. Predicting Early Warning Signs of Psychotic Relapse From Passive Sensing Data: An Approach Using Encoder-Decoder Neural Networks. JMIR Mhealth Uhealth. 2020;8: e19962. doi:10.2196/19962

[3] Tseng VW-S, Sano A, Ben-Zeev D, Brian R, Campbell AT, Hauser M, et al. Using behavioral rhythms and multi-task learning to predict fine-grained symptoms of schizophrenia. Scientific Reports. 2020;10: 15100. doi:10.1038/s41598-020-71689-1

[4] 1. Adler DA, Tseng VW-S, Qi G, Scarpa J, Sen S, Choudhury T. Identifying Mobile Sensing Indicators of Stress-Resilience. Proc ACM Interact Mob Wearable Ubiquitous Technol. 2021;5: 51:1-51:32. doi:10.1145/3463528

We then describe the specific methods used in our analysis in the following subsections (beginning on the original manuscript, page 10), which detail how the sensing data and symptom measures aligned across studies, the model training and validation procedures, as well as the oversampling and personalization procedures we experimented with. These details are included in the following subsections and tables.

Sensor-EMA alignment across studies

Model training and validation

Oversampling to reduce class imbalance

Personalizing models by aligning feature distributions 

Table 4 (Table 2 in the original manuscript)

COMMENT 1B. The sizes of the datasets are really small. In such case, whether the datasets are sufficiently representative for the different mental health symptoms, is not clear. What are the different symptoms reported in the datasets? How many instances of them are reported? How were the feature values set for those features that did not align accross the heterogeneous datasets? It would be interesting to look at the number of instances and size of their feature vectors used for training and testing the model.

RESPONSE 1B: Thank you to the reviewer for this comment. We have adjusted the manuscript as follows, based upon each reviewer question (RQ):

RQ: “What are the different symptoms reported in the datasets?” We have added Tables 2-3 describing the mental health symptoms collected within each study. Please note that for the StudentLife dataset, over 80 different EMAs were collected over the duration of the study. We only include in the manuscript, for brevity, the mental health-related EMA questions, and point readers in the Table legend to the StudentLife data website if they wish to learn more about the full set of EMAs.

RQ: “How many instances of them are reported?” In the manuscript, “Results”, “Aligned data overview” subsection, we describe the EMAs that qualitatively corresponded to the same symptom outcome, as well as the number of instances collected of each EMA within the CrossCheck and StudentLife studies: 

“CrossCheck participants consistently reported 10 EMAs every 2-3 days, resulting in 5,853 total EMAs collected. Applying the ≥30 EMA validation criteria, 5,665 EMAs were collected across 51 participants, median interquartile range (IQR) of 124 (80-141) responses per-participant. Only 3 EMAs - sleep quality, stress, and calmness - had similar question-response structure across studies. 1,079 sleep and 902 stress StudentLife EMAs were collected. 15 StudentLife participants self-reported ≥30 sleep EMAs (597 total, median [IQR] 38 [33-48]), and 9 ≥30 stress EMAs (307 total responses, median [IQR] 33 [32-36]). The calm EMA had <30 responses collected across all StudentLife participants, and was not used.” 

This information - the instances of the sleep and stress EMAs, which were the only two EMAs predicted - are summarized in Table 5 (originally Table 3). The table describes the total number of training instances, and leave-one-subject-out cross-validation instances per sleep and stress EMA across datasets. Table 5 also includes detailed information on the response distribution to the EMA questions, including the median (IQR) of the EMA responses, and the percentage of severe sleep/stress EMA responses (see Figure legend for severe symptom thresholding). We have also moved a supplementary data figure to the main manuscript (Fig 3) to show the specific EMA response distribution breakdown across datasets.

RQ: “How were the feature values set for those features that did not align accross the heterogeneous datasets?” In this study, only sensor features that aligned across datasets were used for prediction. The aligned and misaligned sensor features, with reasoning on this misalignment, are described in Table 4 (originally Table 2) and the “Sensor-EMA alignment across studies” methods subsection. We have added an additional figure (Fig 2 in the revised manuscript) to clarify the 44 aligned features (“size of the feature vector”) used in the analysis, and moved a supplementary figure (now Fig 4) to the main manuscript to show example feature differences across datasets for an example 11 of the 44 features. We also now state the following in the Limitations section, specifically pointing out misaligned sensor features were removed and future work could explore more complex modeling techniques to use both aligned and misaligned features.

“A number of potentially useful data types (audio amplitude, bluetooth, call/text logs, light, phone charge, WiFi location) were misaligned across datasets, and not included as features. Future work can explore more complex modeling techniques to include both aligned and misaligned features across datasets for prediction.”

COMMENT 1C. A claim made in the paper is that the model performs better by combining multiple heterogenous datasets. There are two possibilities for the better results obtained in the experiments. First, the authors are correct. In this case, two small heterogeneous datasets combined into one small (still small, but larger than the individual datasets) should perform better than one large homogeneous (single) dataset. Second, it could be just because of the larger dataset size of the combination of two datasets. So the combined data is just having more samples and thus more representative. It is a well established fact that machine learning models will perform better with more representative samples. From the available datasets and results it is difficult to conclude, which one of the two possibilities is true. To my understanding, I think the second possiblity is more likely to be true. The authors may want to look into it.

RESPONSE 1C: Thank you to the reviewer for this thoughtful comment. It was not our intention to claim the first (two heterogeneous datasets combined perform better than one homogeneous dataset) possibility described. We have edited the introduction of the manuscript to clarify this:

“There is a critical gap in the literature to understand if machine learning models trained using heterogeneous datasets containing distinct populations, collected at different time periods, and with different data collection devices and systems, generalize - i.e. models trained using combined retrospective data to predict held-out participants’ mental health symptoms across multiple studies achieve similar [deleted “or improved”] performance compared to models trained using data collected exclusively from each individual study.” 

We believe our results are more in line with the reviewers’ statement: “it could be just because of the larger dataset size of the combination of two datasets.”, and we have explicitly stated this in the revised manuscript:

“This research does not imply that combining heterogeneous data improves model performance compared to training a machine learning model on a larger homogenous sample. In fact, this research implies the opposite. The decrease in PAD after combining datasets implies that the larger combined training data used in this paper was more representative of out-of-sample participants. Researchers should continue to test models across more diverse datasets to understand when combining data improves or degrades model performance. Model performance can degrade if the combined populations are too dissimilar - known as negative transfer in the machine learning literature [62].”

Our reasoning follows. In the paper, the only empirical evidence we offer as to why performance improved was through the analysis of the PAD (see results “Combined data improves model performance if feature distribution alignment increases”), which actually supports the reviewers’ second claim (“more representative samples”) more than the first claim. If we reduce the PAD between a held-out validation participant and training data through combining datasets, the training data is more representative of the held-out participant since there is greater behavioral feature alignment, and our analysis showed this would lead to an expected model performance improvement.

We’d like to note that combining datasets does not always imply model performance will improve. There is evidence suggesting that model performance can degrade - called negative transfer - if the combined data are too dissimilar. This concept is explored in the paper below:

[1] Rosenstein MT, Marx Z, Kaelbling LP, Dietterich TG. To Transfer or Not To Transfer. : 4.

COMMENT 1D. Some studies use machine/deep learning on large-scale social media data along with other data types to predict mental health issues. The authors should refer some literature on machine learning research for mental health in the paper.

[1] Su, C., Xu, Z., Pathak, J. et al. Deep learning in mental health outcome research: a scoping review. Transl Psychiatry 10, 116 (2020)

[2] S. Ghosh and T. Anwar, Depression Intensity Estimation via Social Media: A Deep Learning Approach, in IEEE Transactions on Computational Social Systems, vol. 8, no. 6, pp. 1465-1474, Dec. 2021

RESPONSE 1D: Thank you for this suggestion. We have added the following sentence in the Introduction to mention machine learning research in mental health broadly, which includes the referenced as well as additional citations.

“In addition to mobile sensing data, researchers have explored using brain images, neural activity recordings, electronic health records, voice and video recordings, and social media data to predict mental health outcomes [8–13].”

Reviewer #2: This article offers a preliminary investigation to determine if models built using combined longitudinal study data to predict mental health symptoms generalise across current publically accessible data. The topic is up-to-date and of major interest to the majority of the journal audience. The paper is well organized and easy to follow. However, the authors need to provide a few details and to overcome a few shortcomings, which can make the paper better. The required shortcomings are given below.

COMMENT 2A. I am unable to verify whether/how the authors plan to make the created data available. The dataset, if not made available for the research community, will reduce the contribution of this work significantly.

RESPONSE 2A: Thank you for the comment, and we agree with the reviewer that the created data is extremely important. In our original submission and revision, we included the following data and code availability statement, including a link where readers can download the two raw datasets used in this work, as well as a GitHub repository containing all code used for analysis. We have sent an inquiry to the editorial office to confirm the reviewer has access to the data availability statement, and copied the statement below for the reviewer’s convenience:

“The two raw datasets used in this work can be found on the Precision Behavioral Health Initiative @ Cornell Tech’s website (https://pbh.tech.cornell.edu/data.html). We created a public repository (https://github.com/CornellPACLab/data_heterogeneity) with analysis code. All code written for this study was built using open source Python packages, including sci-kit learn for machine learning, statsmodels for linear modeling, and pingouin for statistical testing.”

COMMENT 2B. Figures in this paper have low quality.

RESPONSE 2B: We appreciate the feedback. We have confirmed that we adhered to PLOS ONE’s policy of requiring “.tiff” files at >=300 pixels per inch resolution in our original submission and revision, as per the PLOS ONE figure guidelines (https://journals.plos.org/plosone/s/figures). We understand the PDF figures are low quality due to the PLOS ONE’s compression of figures in the submission. 

To the best of our knowledge, the PLOS ONE submitted manuscript includes a link on the top-right hand corner of each low-quality Figure page in the PDF that states “Click here to access/download;Figure;FigX.tiff”. By clicking those links, the Reviewer can download a higher quality figure. We have reached out to the editorial office to confirm that the reviewer has access to the higher quality figures, and hope the reviewer is able to work with the editorial office to access the higher quality figures for their review.

COMMENT 2C. The overall language quality could be improved. The structure of some sentences makes the reading really hard.

RESPONSE 2C: We have revised the manuscript to improve the sentence structure, and we hope we were able to modify the specific sections of the manuscript the reviewer is referring to.

COMMENT 2D. Some related studies could be discussed.

a. Ghosh, Soumitra, Asif Ekbal, and Pushpak Bhattacharyya. "What Does Your Bio Say? Inferring Twitter Users' Depression Status From Multimodal Profile Information Using Deep Learning." IEEE Transactions on Computational Social Systems (2021).

b. Ghosh, Soumitra, Asif Ekbal, and Pushpak Bhattacharyya. "Cease, a corpus of emotion annotated suicide notes in english." Proceedings of The 12th Language Resources and Evaluation Conference. 2020.

c. Ghosh, Soumitra, Asif Ekbal, and Pushpak Bhattacharyya. "A multitask framework to detect depression, sentiment and multi-label emotion from suicide notes." Cognitive Computation 14.1 (2022): 110-129.

RESPONSE 2D: Thank you to the reviewer for these suggestions. Given our paper is focused on predicting mental health, and not necessarily focused on data corpuses, we have cited the suggested sources [a] and [c], but not [b]. These additions are found in the Introduction, in the sentence copied below:

“In addition to mobile sensing data, researchers have explored using brain images, neural activity recordings, electronic health records, voice and video recordings, and social media data to predict mental health outcomes [8–13].”

COMMENT 2E. Some more references in 2020 and 2021 should be considered.

RESPONSE 2E: Our revision includes the following references published in 2020, 2021, and 2022. Please note the numbering in the list below does not correspond to the reference order in the revised or original manuscript.

[1] Adler DA, Ben-Zeev D, Tseng VW-S, Kane JM, Brian R, Campbell AT, et al. Predicting Early Warning Signs of Psychotic Relapse From Passive Sensing Data: An Approach Using Encoder-Decoder Neural Networks. JMIR MHealth UHealth. 2020;8: e19962. doi:10.2196/19962

[2] Henson P, D’Mello R, Vaidyam A, Keshavan M, Torous J. Anomaly detection to predict relapse risk in schizophrenia. Transl Psychiatry. 2021;11: 1–6. doi:10.1038/s41398-020-01123-7

[3] Bardram JE, Matic A. A Decade of Ubiquitous Computing Research in Mental Health. IEEE Pervasive Comput. 2020;19: 62–72. doi:10.1109/MPRV.2019.2925338

[4] Sultana M, Al-Jefri M, Lee J. Using Machine Learning and Smartphone and Smartwatch Data to Detect Emotional States and Transitions: Exploratory Study. JMIR MHealth UHealth. 2020;8: e17818. doi:10.2196/17818

[5] Opoku Asare K, Terhorst Y, Vega J, Peltonen E, Lagerspetz E, Ferreira D. Predicting Depression From Smartphone Behavioral Markers Using Machine Learning Methods, Hyperparameter Optimization, and Feature Importance Analysis: Exploratory Study. JMIR MHealth UHealth. 2021;9: e26540. doi:10.2196/26540

[6] Sükei E, Norbury A, Perez-Rodriguez MM, Olmos PM, Artés A. Predicting Emotional States Using Behavioral Markers Derived From Passively Sensed Data: Data-Driven Machine Learning Approach. JMIR MHealth UHealth. 2021;9: e24465. doi:10.2196/24465

[7] Zhang Y, Folarin AA, Sun S, Cummins N, Ranjan Y, Rashid Z, et al. Predicting Depressive Symptom Severity Through Individuals’ Nearby Bluetooth Device Count Data Collected by Mobile Phones: Preliminary Longitudinal Study. JMIR MHealth UHealth. 2021;9: e29840. doi:10.2196/29840

[8] Shaukat-Jali R, Zalk N van, Boyle DE. Detecting Subclinical Social Anxiety Using Physiological Data From a Wrist-Worn Wearable: Small-Scale Feasibility Study. JMIR Form Res. 2021;5: e32656. doi:10.2196/32656

[9] Haines-Delmont A, Chahal G, Bruen AJ, Wall A, Khan CT, Sadashiv R, et al. Testing Suicide Risk Prediction Algorithms Using Phone Measurements With Patients in Acute Mental Health Settings: Feasibility Study. JMIR MHealth UHealth. 2020;8: e15901. doi:10.2196/15901

[10] Müller SR, Chen X (Leslie), Peters H, Chaintreau A, Matz SC. Depression predictions from GPS-based mobility do not generalize well to large demographically heterogeneous samples. Sci Rep. 2021;11: 14007. doi:10.1038/s41598-021-93087-x

[11] Adler DA, Tseng VW-S, Qi G, Scarpa J, Sen S, Choudhury T. Identifying Mobile Sensing Indicators of Stress-Resilience. Proc ACM Interact Mob Wearable Ubiquitous Technol. 2021;5: 51:1-51:32. doi:10.1145/3463528

[12] Tseng VW-S, Sano A, Ben-Zeev D, Brian R, Campbell AT, Hauser M, et al. Using behavioral rhythms and multi-task learning to predict fine-grained symptoms of schizophrenia. Sci Rep. 2020;10: 15100. doi:10.1038/s41598-020-71689-1

[13] Cooper AF, Lu Y, Forde JZ, De Sa C. Hyperparameter Optimization Is Deceiving Us, and How to Stop It. ArXiv210203034 Cs. 2021 [cited 4 Nov 2021]. Available: http://arxiv.org/abs/2102.03034

[14] Farber DG, Kemmer DD. Common Measures for Mental Health Science Laying the Foundations June 2020. : 3.

[15] Strubell E, Ganesh A, McCallum A. Energy and Policy Considerations for Modern Deep Learning Research. Proc AAAI Conf Artif Intell. 2020;34: 13693–13696. doi:10.1609/aaai.v34i09.7123

[16] Su C, Xu Z, Pathak J, Wang F. Deep learning in mental health outcome research: a scoping review. Transl Psychiatry. 2020;10: 1–26. doi:10.1038/s41398-020-0780-3

[17] Ghosh S, Ekbal A, Bhattacharyya P. What Does Your Bio Say? Inferring Twitter Users’ Depression Status From Multimodal Profile Information Using Deep Learning. IEEE Trans Comput Soc Syst. 2021; 1–11. doi:10.1109/TCSS.2021.3116242

[18] Ghosh S, Anwar T. Depression Intensity Estimation via Social Media: A Deep Learning Approach. IEEE Trans Comput Soc Syst. 2021;8: 1465–1474. doi:10.1109/TCSS.2021.3084154

[19] Ghosh S, Ekbal A, Bhattacharyya P. A Multitask Framework to Detect Depression, Sentiment and Multi-label Emotion from Suicide Notes. Cogn Comput. 2022;14: 110–129. doi:10.1007/s12559-021-09828-7

[20] Adler DA, Wang F, Mohr DC, Estrin D, Livesey C, Choudhury T. A call for open data to develop mental health digital biomarkers. BJPsych Open. 2022;8. doi:10.1192/bjo.2022.28

---

## [Decision Letter · Decision Letter 1]

23 Mar 2022

Machine learning for passive mental health symptom prediction: generalization across different longitudinal mobile sensing studies

PONE-D-22-01484R1

Dear Dr. Adler,

We’re pleased to inform you that your manuscript has been judged scientifically suitable for publication and will be formally accepted for publication once it meets all outstanding technical requirements.

Kind regards,

Chi-Hua Chen, Ph.D.

Academic Editor

PLOS ONE

Additional Editor Comments (optional):

Reviewers' comments:

Reviewer's Responses to Questions

**Comments to the Author**

1. If the authors have adequately addressed your comments raised in a previous round of review and you feel that this manuscript is now acceptable for publication, you may indicate that here to bypass the “Comments to the Author” section, enter your conflict of interest statement in the “Confidential to Editor” section, and submit your "Accept" recommendation.

Reviewer #1: All comments have been addressed

Reviewer #2: All comments have been addressed

2. Is the manuscript technically sound, and do the data support the conclusions?

Reviewer #1: Yes

Reviewer #2: Yes

3. Has the statistical analysis been performed appropriately and rigorously? 

Reviewer #1: Yes

Reviewer #2: Yes

4. Have the authors made all data underlying the findings in their manuscript fully available?

Reviewer #1: Yes

Reviewer #2: Yes

5. Is the manuscript presented in an intelligible fashion and written in standard English?

Reviewer #1: Yes

Reviewer #2: Yes

6. Review Comments to the Author

Reviewer #1: The authors have addressed all my comments. The paper has some interesting analysis worth publishing. It may be accepted now.

Reviewer #2: The authors have addressed essentially all my previous concerns and their revisions have substantially improved the manuscript.

7. PLOS authors have the option to publish the peer review history of their article (what does this mean?). If published, this will include your full peer review and any attached files.

Reviewer #1: No

Reviewer #2: No

---

## [Editor Report · Acceptance letter]

19 Apr 2022

PONE-D-22-01484R1 

Machine learning for passive mental health symptom prediction: generalization across different longitudinal mobile sensing studies 

Dear Dr. Adler:

I'm pleased to inform you that your manuscript has been deemed suitable for publication in PLOS ONE. Congratulations! Your manuscript is now with our production department. 

Kind regards, 

on behalf of

Professor Chi-Hua Chen 

Academic Editor

PLOS ONE